# Oncological Outcomes and Patterns of Recurrence after the Surgical Resection of an Invasive Intraductal Papillary Mucinous Neoplasm versus Primary Pancreatic Ductal Adenocarcinoma: An Analysis from the German Cancer Registry Group of the Society of German Tumor Centers

**DOI:** 10.3390/cancers16112016

**Published:** 2024-05-26

**Authors:** Thaer S. A. Abdalla, Jannis Duhn, Monika Klinkhammer-Schalke, Sylke Ruth Zeissig, Kees Kleihues-van Tol, Kim C. Honselmann, Rüdiger Braun, Markus Kist, Louisa Bolm, Lennart von Fritsch, Hryhoriy Lapshyn, Stanislav Litkevych, Richard Hummel, Sergii Zemskov, Ulrich Friedrich Wellner, Tobias Keck, Steffen Deichmann

**Affiliations:** 1Department of Surgery, University Medical Center Schleswig-Holstein, Campus Lübeck, 23562 Lübeck, Germany; 2Network for Care, Quality and Research in Oncology (ADT), German Cancer Registry Group of the Society of German Tumor Centers, 14057 Berlin, Germany; 3Institute of Clinical Epidemiology and Biometry (ICE-B), University of Würzburg, 97974 Würzburg, Germany; 4Department of Surgery, University Medical Center Greiswald, 17489 Greifswald, Germany; 5Department of General Surgery, Bogomolets National Medical University, 01601 Kiev, Ukraine

**Keywords:** IPMN, PDAC, patterns of recurrence, pancreatic cancer

## Abstract

**Simple Summary:**

This retrospective study, based on data from the German Cancer Registry Group of the Society of German Tumor Centers, analyzed a specific type of malignant cystic pancreatic neoplasm, called invasive IPMN, and compared it to pancreatic ductal adenocarcinoma (PDAC). Invasive IPMN tumors were of smaller size and less aggressive, and their complete removal during surgery was achieved more frequently. Patients with invasive IPMN generally had better outcomes after surgery, with longer survival times and fewer recurrences, especially in the early stages of the disease. Interestingly, chemotherapy after surgery did not result in a survival benefit in patients with invasive IPMN. Overall, this retrospective study suggests that invasive IPMN represents a less aggressive type of pancreatic cancer resulting in a favorable prognosis, especially in early tumor stages, thus highlighting the relevance of the already existing surveillance programs of this entity.

**Abstract:**

Background: Intraductal papillary mucinous neoplasms (IPMNs) are premalignant cystic neoplasms of the pancreas (CNPs), which can progress to invasive IPMN and pancreatic cancer. The available literature has shown controversial results regarding prognosis and clinical outcomes after the resection of invasive IPMN. Aims: This study aims to characterize the oncologic outcomes and metastatic progression pattern after the resection of non-metastatic invasive IPMN. Methods: Data were obtained from 24 clinical cancer registries participating in the German Cancer Registry Group of the Society of German Tumor Centers (ADT). Patients with invasive IPMN (*n* = 217) as well as PDAC (*n* = 5794) between 2000 and 2021 were included and compared regarding oncological outcomes. Results: Invasive IPMN was significantly smaller in size (*p* < 0.001) and of a lower tumor grade (*p* < 0.001), with fewer lymph node metastases (*p* < 0.001), lymphangiosis (*p* < 0.001), and consequently a higher R0 resection rate (88 vs. 74%) compared to PDAC. Moreover, invasive IPMN was associated with fewer local (11 vs. 15%) and distant recurrences (29 vs. 46%) and metastasized more frequently in the lungs only (26% vs. 14%). Invasive IPMN was associated with a longer median OS (29 vs. 19 months) and DFS (31 vs. 15 months) compared to PDAC and stayed independently prognostic in multivariable analyses. These survival differences were most pronounced in early tumor stages. Interestingly, postoperative chemotherapy was not associated with improved overall survival in surgically resected invasive IPMN. Conclusions: Invasive IPMN is a rare pancreatic entity with increasing incidence in Germany. It is associated with favorable histopathological features at the time of resection and longer OS and DFS compared to PDAC, particularly before the locoregional spread has occurred. Invasive IPMNs are associated with lung-only metastasis. The benefit of postoperative chemotherapy after the resection of invasive IPMN remains uncertain.

## 1. Introduction

The first description of intraductal papillary mucinous neoplasms (IPMNs) can be traced back to the 1980s, when Ohashi et al. described four cases of mucin-producing tumors with papillary growth and ductal dilatation [1]. The recognition of IPMN as a unique entity revolutionized the understanding of pancreatic pathology and provided new insights into the disease progression and management of these neoplasms. Subsequently, in 2000, the World Health Organization (WHO) recognized the significance of IPMNs and included them in the classification of pancreatic tumors [2]. In 2016, further guidelines for pathological tumor stage classification in resected IPMNs were introduced [3].

IPMNs arise from the pancreatic ductal epithelium and can be morphologically differentiated into the main duct, side branch, and mixed-type IPMNs, according to European guidelines [4]. IPMNs can progress to high-grade dysplasia (HGD) and invasive carcinoma; however, clinical decision-making processes remains complex. Currently, surgical resection is indicated in the presence of certain high-risk stigmata, such as jaundice, enhancing mural nodules ≥ 5 mm, the presence of solid components, positive cytology for HGD or invasive cancer, and dilation of the pancreatic duct ≥ 10 mm [3,5]. The European guidelines further define “worrisome features”, representing relative indications for surgery. Alternatively, patients with IPMN can be eligible for clinical surveillance using MRI and/or endoscopic ultrasound (EUS) technologies [4]. Thereby, the surveillance intervals are dependent on the presence or absence of these worrisome features [3,5].

IPMN with associated invasive carcinoma (invasive IPMN) is currently treated in an analogous manner to primary pancreatic ductal adenocarcinoma (PDAC) [6]. However, two meta-analyses have shown significant advantages in the overall survival of patients suffering from invasive IPMN compared to PDAC [7,8]. A recent analysis from the US National Cancer Database showed that invasive IPMN is frequently diagnosed at earlier stages, leading to significantly higher proportions of patients eligible for surgical resection. In a study that conducted uni- and multivariate regression analyses, primarily resectable invasive IPMN showed a significantly higher proportion of negative resection margins and better overall survival compared to PDAC [9]. However, a large retrospective analysis from the Karolinska Institute demonstrated that the overall survival of invasive IPMN is non-superior to PDAC in higher tumor stages (≥pT2, >pN1, and ≥cM1) [10].

While the recurrence of upfront resected PDAC has been described in detail [11,12], the recurrence patterns of primarily resected invasive IPMNs and their impact on long-term survival are still ill-defined.

In this study, we provide a large-scale analysis of the oncologic outcomes and recurrence pattern of resected non-metastatic invasive IPMN compared to primary PDAC using population-based data from clinical cancer registries in Germany.

## 2. Materials and Methods

This retrospective study was performed using data from 24 population-based clinical cancer registries participating in the German Cancer Registry Group of the Society of German Tumor Centers (ADT). The anonymized data are included in a large-scale dataset by the ADT, available for analysis by certified centers. The registry data were used according to the regulations of the ADT. The study was approved by the ethics committee of the University of Lübeck, Germany (#2023-156). 

Of all patients diagnosed with pancreatic malignancy (codes C25.0-C25.9) in 2000–2021, patients suffering from primary PDAC were separated from those suffering from invasive IPMN based on ICD-O 3. edition morphology code (ICD-O-3) (not otherwise specified PDAC (ICD-O-3 morphology code 8500/3); invasive IPMN (ICD-O-3 morphology code 8453/3)) [13]. Additionally, the analysis was limited to patients without distant metastases (M0) who underwent upfront surgery and for whom data on overall survival (OS), T-stage, lymph node status, and resection margin status were available in the dataset.

The following parameters were retrieved from the cancer registry data: sex, age at diagnosis (years), lymph node metastases (N0 and N+), T-stage (T3–T4), lymphangiosis (L0 and L1), hemangiosis (V0 and V1), tumor grade (G1–G3), resection margin status (R0, R1, and R2), tumor location (pancreatic head, body, and tail), operation type (pancreatoduodenectomy, distal pancreatectomy, and total pancreatectomy), follow-up time (months after diagnosis), and status at last follow-up (dead/alive and disease recurrence status).

The variables of age, lymph node metastasis, tumor location, and resection status were dichotomized as followed: age ≤ 65 years versus > 65 years, lymph node metastasis N0 vs. N+, tumor location (head vs. tail/body), and resection margins being negative (R0) versus positive (R1/R2). The TNM classification changed over the selected study period (2000–2021), mainly by the update of the UICC/AJCC classification in 2016. The main change was regarding the T-stage in T3 and T4 tumors, where the definition differed according to extra-pancreatic organ involvement. To retrospectively compensate for this effect, a sensitivity analysis was performed for the time of diagnosis (2000–2015 vs. 2016–2021) (Appendix A). As the metric tumor size and the exact tumor extension were not available in our dataset, restaging according to the current TNM classification was only possible by combining T-stages T3 and T4 into one group, therefore including all patients regardless of organ involvement [14,15,16].

### Statistical Methods

In the context of statistical analysis, SPSS 26 for Windows (Armonk, NY, USA) was employed. The descriptive statistics encompassed absolute and relative frequencies, median, interquartile range, and Kaplan–Meier estimates with the corresponding plots. The overall survival was computed as the duration from the date of diagnosis to the date of death, while disease-free survival was defined as the interval from tumor resection to the occurrence of local or metastatic recurrence. Statistical examinations involved the application of the chi-squared test, log-rank test, and uni- and multivariable Cox regression models. A two-sided significance level set at *p* < 0.05 was assumed as statistical significance throughout the study.

## 3. Results

### 3.1. Patient Cohort Characteristics

A total of 5794 patients with primary PDAC and 217 patients with invasive IPMN met the inclusion and exclusion criteria. The frequency of diagnoses of both tumor entities increased over the years (Appendix A). 

Both PDAC (51.8%) and invasive IPMN (53.5%) occurred slightly more often in men. The median age (± IQR) at diagnosis was 70 ± 14 years for PDAC and 71 ± 13 years for invasive IPMN, and invasive IPMN occurred significantly more often in patients aged 65 years or older (*p* = 0.043). In addition, tumors categorized as invasive IPMN were significantly smaller in size (*p* < 0.001) and of a lower tumor grade (*p* < 0.001, Table 1). Furthermore, invasive IPMN was associated with less lymph node metastases (Figure 1, 41% vs. 69%, *p* < 0.001), lymphangiosis (46% vs. 58%, *p* < 0.001), and hemangiosis (18% vs. 26%, *p* = 0.015). 

The performed surgical procedure also differed significantly between the two entities (*p* < 0.001). The patients with PDAC were more prone to undergo pancreaticoduodenectomy (PD) compared to those with invasive IPMN. On the other hand, invasive IPMN more often required total pancreatectomies (TP) compared to PDAC (22% vs. 12%). The R0 resection rate was significantly higher in patients with invasive IPMN (88% vs. 74%, *p* < 0.001).

Importantly, both local recurrence (11% vs. 15%, *p* = 0.011) and metachronous distant metastases (29% vs. 46%, *p* < 0.001) occurred less frequently in patients with invasive IPMN compared to PDAC (Table 1). 

### 3.2. Invasive IPMN Biology Is Associated with Better Overall and Disease-Free Survival

Invasive IPMN biology was associated with a significantly better overall survival in the patients that received upfront resection (Figure 2A) compared to PDAC (median OS 29 ± 6 vs. 19 ± 1 months, *p* < 0.001; respectively). The 1-, 3-, and 5-year survivals of non-metastatic invasive IPMN were 78%, 43%, and 33% compared to 66%, 22%, and 10% in PDAC (Appendix A).

Information regarding disease-free survival (DFS) was available for 4461 (4277 PDAC and 184 invasive IPMN) patients (Appendix A). In these cases, invasive IPMN was associated with a favorable DFS compared to PDAC (median DFS 31 ± 18.0 vs. 15 ± 1 months, *p* < 0.001).

We further included a subgroup analysis, comparing the OS and DFS of invasive IPMN and PDAC within the distinct TN stages. Invasive IPMN in small tumor stages (T1-2) and in the absence of local lymph node metastases (Figure 2B) showed a significantly improved OS compared to PDAC of the same tumor stage (median OS 64.2 ± 21.4 vs. 23.2 ± 4.0 months, *p* < 0.001). Interestingly, the OS was also improved in pT3-4 N+ tumors (median OS 20 ± 4 vs. 16 ± 1 months, *p* = 0.017). On the other hand, we did not observe significant differences for pT1-2 N+ and pT3-4 N0 tumors between both groups (Figure 2B–E). Furthermore, invasive IPMN also showed a significantly improved DFS over PDAC in small, localized tumors (T1-T2 N0), which was not observed in other tumor stages (Appendix A–E). 

### 3.3. IPMN Biology Is an Independent, Positive Prognostic Factor

The multivariable regression analysis for the overall survival revealed that the increase in tumor size T2 vs. T1 (HR: 1.58, CI 95%: 1.32–1.91, *p* < 0.001), T3 vs. T1 (HR: 1.53, CI 95%: 1.28–1.83, *p* < 0.001), the presence of lymph node metastasis (HR: 1.41, CI 95% = 1.31–1.51, *p* < 0.001), as well as positive resection margins (HR: 1.46, CI 95% = 1.36–1.56, *p* < 0.001) were independent, negative prognostic factors for overall survival (Table 2). 

Moreover, the histological evidence of IPMN was an independent factor for a longer overall survival (HR: 0.67, CI 95%: 0.55–0.82, *p* < 0.001) compared to invasive PDAC. However, the surgical resection margins remained the only potentially modifiable risk factor in this analysis.

### 3.4. Metachronous Metastasis Locations Differ between PDAC and Invasive IPMN

Invasive IPMN and PDAC showed different patterns of recurrence regarding the locations of metachronous metastases (Figure 3A). In the overall study cohort, the incidence of multiple site metastasis, among all patients with metachronous metastases, tended to be lower in invasive IPMN compared to PDAC (14.0% vs. 22.2%). The proportion of patients showing liver-only metastases was comparable in each group (42% vs. 40%). Peritoneal-only metastases occurred slightly more often in PDAC (12% vs. 18%), whereas lung-only metastases were more common in invasive IPMN (26% vs. 14%) (Appendix A).

In a subgroup analysis, early-stage invasive IPMN (pT1) tended to metastasize to the lungs only (75% (3/4) vs. 14.3% (7/49)) and did not recur in multiple locations as observed in PDAC (0% vs. 24.5% (12/49)). In more advanced stages of invasive IPMN, the recurrence showed a similar pattern to that of PDAC (Figure 3B,C).

### 3.5. Effect of Postoperative Chemotherapy on Invasive IPMN and PDAC

Data on postoperative chemotherapy were reported in 186 patients with resected non-metastatic invasive IPMN (85.7%). Of these, 112 patients did not receive chemotherapy (60.2%), 11 patients received the FOLFIRINOX or FOLFIRI regimen (5.9%), 55 patients received gemcitabine-based chemotherapy (29.6%), and 8 patients were administered 5-FU monotherapy (4.3%). Finally, we compared the impact of postoperative chemotherapy—regardless of the used protocol—on the overall survival of patients with invasive IPMN or PDAC (Figure 4A,B). In this case, the postoperative chemotherapy did not significantly improve the overall survival of patients (median OS: 28.9 ± 6.1 vs. 23.8 ± 11.6 months, *p* = 0.089). In comparison, the patients with PDAC benefited from postoperative chemotherapy (median OS: 21.8 ± 0.9 vs. 11.8 ± 0.7, *p* < 0.001).

## 4. Discussion

We present the first population-based study from Germany regarding oncological outcomes after the resection of invasive IPMN compared to PDAC. Our results suggest that invasive IPMN biology is associated with a longer overall survival, less aggressive histopathological features, and different patterns of recurrence and metastasis compared to PDAC.

IPMN can progress, as part of its natural course, into high-grade dysplasia and eventually into invasive carcinoma. Nevertheless, invasive IPMN apparently harbors a more indolent biology and favorable survival compared to PDAC. 

In our cohort, non-metastatic invasive IPMN was associated with a longer median overall survival compared to non-metastatic PDAC (29 vs. 19 months, *p* < 0.001). The 1-,3-, and 5-year survivals of non-metastatic invasive IPMN were 78%, 43%, and 33% compared to 66%, 22%, and 10% in PDAC. This advantage in overall survival for invasive IPMN was also present in the multivariable analysis, independent of age, sex, tumor size, lymph node metastasis, and resection margin status (HR: 0.67, CI 95%: 0.55–0.82, *p* < 0.001). 

Interestingly, when stratifying patients according to the T-stage and nodal involvement, the univariable analysis revealed that invasive IPMN had favorable OS and DFS in T1-2 N0 tumors (*p* < 0.001). Therefore, in the setting of nodal metastasis or large tumors > 4 cm, the prognosis of invasive IPMN is poor and comparable to that of PDAC. These findings are in line with the results of previous publications [8,9,17]. The reason behind the shift in biological behavior cannot be explained using the available dataset. But the results are suggestive that the early pancreatic resection of invasive IPMN improves oncological outcomes. 

Another important aspect is that invasive IPMN was associated with less aggressive histopathological features compared to PDAC. The tumors were smaller in size, more differentiated, and associated with less lymph node metastases, less hemangiosis, and lymphangiosis compared to PDAC. These findings are comparable to the results in the existing literature [8,9,17]. 

As previously suggested, the favorable features of invasive IPMN could result from a selection bias since it must be suspected that many patients were either under routine surveillance or had an accidental diagnosis, thus receiving early intervention and resection compared to PDAC. Similarly, invasive IPMN may cause symptoms early in the course of the disease, leading to prompt radiological detection, unlike PDAC, which remains asymptomatic and presents itself usually after locoregional or distant metastasis has occurred. For instance, Salvia et al. reported that 23% of all invasive IPMN presented with acute pancreatitis [18]. Information regarding the presenting symptoms at diagnosis is not available in the cancer registry dataset and, therefore, could not be validated in this study, representing an important limitation. Nevertheless, the favorable features of invasive IPMN could explain the higher R0 resection rates compared to PDAC (84% vs. 74%), which in turn are associated with a longer OS after complete resection with a negative versus positive resection margin (1-year OS: 80% vs. 64%; 3-year OS: 47% vs. 18%). Another observation is that 22% of all patients with invasive IPMN required total pancreatectomy compared to 12% in PDAC. This could be explained by the multifocal nature of IPMN and the cystic malformation of the pancreas, which might require the completion of pancreatectomy at the time of surgery. This high rate of total pancreatectomies in invasive IPMN was also reported in previous large studies originating from the United States and Sweden [9,19].

Furthermore, we analyzed the patterns of recurrence and metastasis after tumor resection in PDAC and invasive IPMN. Invasive IPMN was associated with a longer DFS (31 vs. 15 months, *p* < 0.001), with fewer incidences of local recurrence (11% vs. 15%, *p* = 0.011) and metachronous distant metastases (27% vs. 46%, *p* < 0.001). Moreover, the distribution of metastases was different between both entities (Figure 2). Metastases in multiple sites were reported in 22% of patients with PDAC compared to 14% of patients with invasive IPMN. The incidence of lung-only metastasis was reported in 26% of the patients with invasive IPMN compared to 14% of patients with PDAC. Oweira et al. previously demonstrated, in an analysis of 13,233 patients with metastatic PDAC using the SEER database, that the presence of lung-only metastasis had a positive impact on survival compared to patients with liver-only metastasis or metastases in multiple locations [20]. Later, Kurreck et al. demonstrated, in a pooled analysis of 912 patients from three adjuvant therapy trials (CONKO-001, CONKO-005, and CONKO-006), that metachronous isolated lung metastasis was associated with a favorable OS and DFS [21]. Furthermore, in a multicentric study from Japan, Homma et al. demonstrated that patients with lung-only metastasis after pancreatectomy had a median OS of 23 months and had favorable outcomes compared to other locations of recurrence [22]. Combining the results of the aforementioned studies with our findings adds further insights into the more favorable biology of invasive IPMN compared to PDAC, which seems to be independent of the timing of disease detection and may explain the observed differences in survival. 

Finally, we conducted a subgroup analysis of the effect of postoperative chemotherapy in invasive IPMN after pancreatic resection. Out of the 217 patients, 112 patients underwent surgery only, and 99 patients received postoperative chemotherapy. The univariable analysis revealed that postoperative chemotherapy did not improve OS in patients with non-metastatic IPMN (*p* = 0.089). On the other hand, postoperative chemotherapy was associated with longer overall survivals in PDAC (*n* = 4256, *p* < 0.001) (Figure 4). This confirms the results of two previous studies, where postoperative chemotherapy did not confer survival benefits in patients with invasive IPMN [23,24]. Although only 46% of patients with invasive IPMN received postoperative chemotherapy, this represents the current clinical practice. In a study conducted by Petruch et al., comparing textbook outcomes in resectable pancreatic cancer between Germany and the US, it was found that only 36% of resected pancreatic cancer patients in Germany and 30% in the US received adjuvant chemotherapy. Hence, while the numbers are relatively low, they are reflective of real-world practices [25].

Several limitations should be considered when interpreting the results of this study, mostly being related to the retrospective design and registry-based analysis. Since our data were derived from 24 clinical cancer registries, inconsistency in reporting likely existed. Nevertheless, the German Cancer Registry Group of the Society of German Tumor Centers (ADT) has implemented training programs aimed at standardizing data input procedures, involving a 20-day training period followed by examination under supervision [26]. A differentiation between preoperative diagnosis and postoperative diagnosis and CRM (−/+) was not available in our dataset. Moreover, the registry data did not include detailed information regarding preoperative morphological features (side branch vs. main-duct IPMN) or the histological subtype (intestinal vs. colloid IPMN), as well as preoperative duct diameter; therefore, our analysis could not be adjusted for potential confounding factors. Moreover, we cannot exclude that some patients with invasive IPMN were reported as PDAC, especially in the initial period of our study. Lastly, the TNM classification has changed over time, especially regarding the T-stage; therefore, we provided a sensitivity analysis from the time periods before and after 2016, while combining the T3-T4 stages to compensate for potential deficits. Despite these drawbacks, our large sample size provides analytic power, particularly considering this rare pancreatic entity. 

## 5. Conclusions

Invasive IPMN is a rare pancreatic entity with increasing incidence in Germany. It is associated with favorable histopathological features at the time of resection and longer OS and DFS compared to PDAC, particularly before locoregional spread has occurred. Invasive IPMNs are associated with lung-only metastasis. The benefit of postoperative chemotherapy after the resection of invasive IPMN remains uncertain.

## Figures and Tables

**Figure 1 cancers-16-02016-f001:**
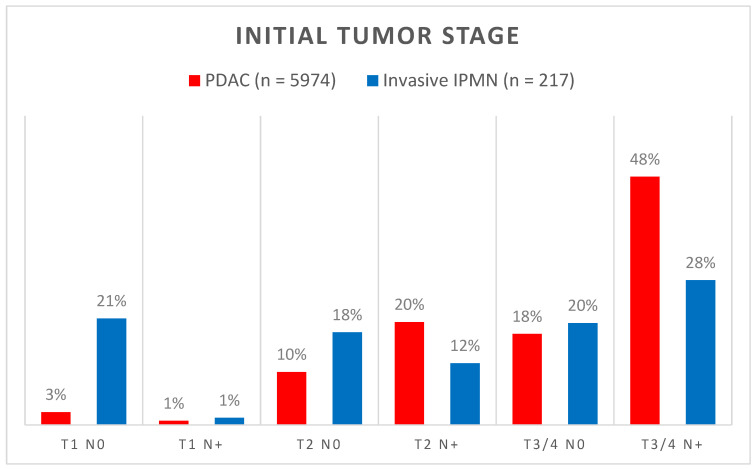
Comparison of tumor stages at diagnosis of primarily resected PDAC (red bars) and invasive IPMN (blue bars). Numbers above the bars indicate the percentage within the respective tumor type.

**Figure 2 cancers-16-02016-f002:**
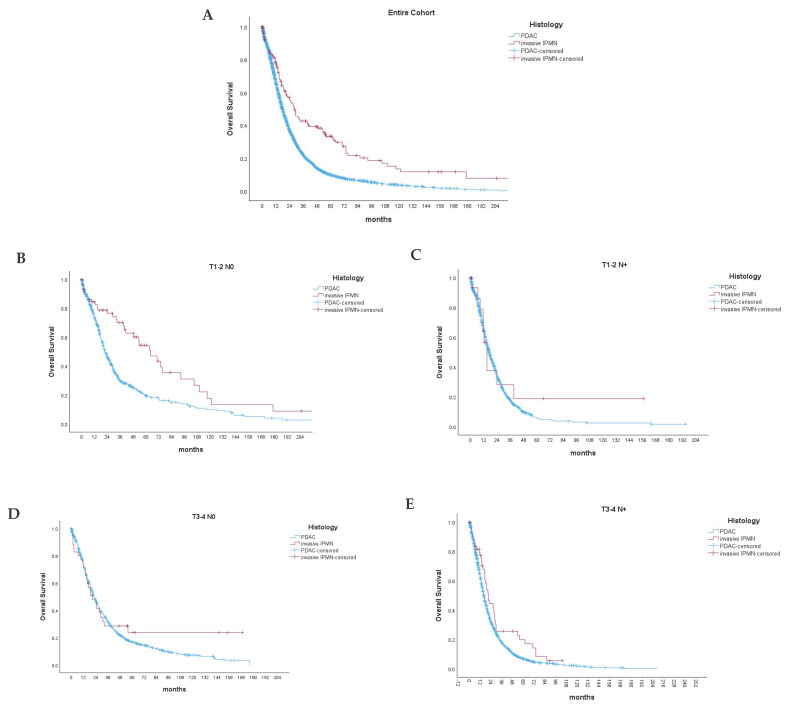
Overall survival (*p* < 0.001) (**A**) of primarily resected PDAC (red line) and invasive IPMN (blue line). Subgroup analyses comparing the overall survival of invasive IPMN and PDAC among different tumor and nodal stages (**B**–**E**). *p* < 0.001 (**B**), *p* = 0.302 (**C**), *p* = 0.187 (**D**), *p* = 0.017 (**E**).

**Figure 3 cancers-16-02016-f003:**
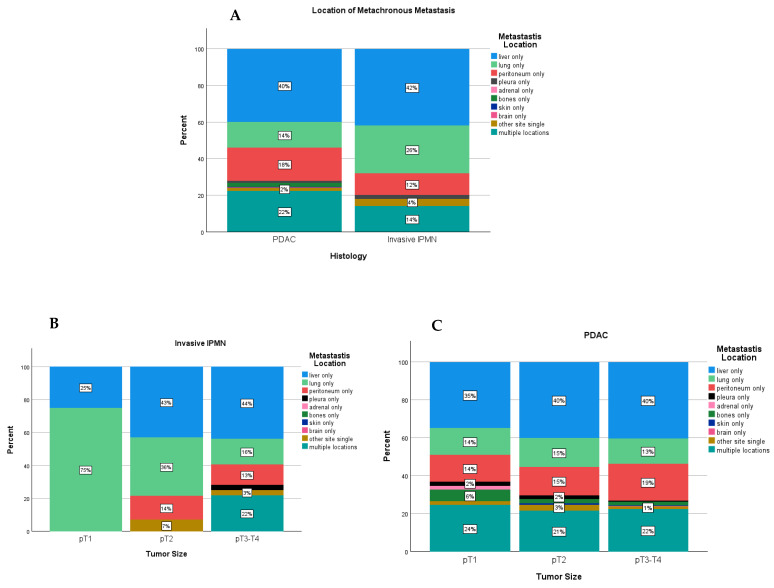
Locations of metachronous metastases in invasive IPMN and PDAC among all patients with distant metastases (**A**) and separated by the initial tumor size (**B**,**C**).

**Figure 4 cancers-16-02016-f004:**
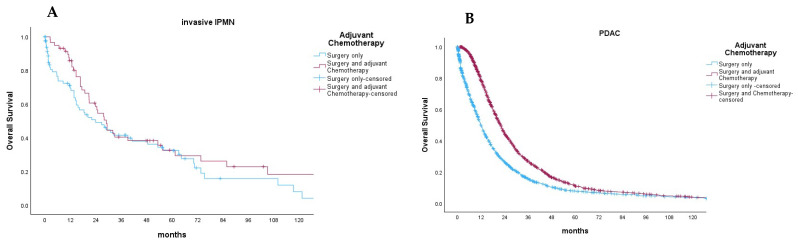
Effect of postoperative chemotherapy (black line) regardless of the regimen compared to surgery only (red line) on the overall survival of patients with invasive IPMN (*p* = 0.089) (**A**) and PDAC (*p* < 0.001) (**B**).

**Table 1 cancers-16-02016-t001:** Characteristics and statistical comparison of the study cohort using the chi-squared test.

	PDAC	Invasive IPMN	*p*-Value
Sex	*n* = 5793	*n* = 217	
Male	3093 (52%)	116 (54%)	
Female	2880 (48%)	101 (46%)	
Age	*n* = 5972	*n* = 217	0.043
<65 years	1902 (32%)	55 (25%)	
≥65 years	4070 (68%)	162 (75%)	
Tumor Size	*n* = 5974	*n* = 217	<0.001
pT1	194 (3%)	48 (22%)	
pT2	1805 (30%)	65 (30%)	
pT3-4	3975 (67%)	104 (48%)	
Lymph Node Metastasis	*n* = 5974	*n* = 217	<0.001
pN0	1845 (31%)	127 (59%)	
pN+	4120 (69%)	90 (41%)	
Tumor Grading	*n* = 5839	*n* = 198	<0.001
G1	265 (4%)	32 (16%)	
G2	3056 (52%)	100 (51%)	
G3	2506 (43%)	63 (32%)	
G4	12 (<1%)	2 (1%)	
Lymphangiosis	*n* = 5530	*n* = 197	<0.001
L0	2311 (42%)	107 (54%)	
L1	3219 (58%)	90 (46%)	
Hemangiosis	*n* = 5386	*n* = 196	0.015
V0	3981 (74%)	160 (82%)	
V1	1405 (26%)	36 (18%)	
Resection Margins	*n* = 5974	*n* = 217	<0.001
R0	4414 (74%)	190 (88%)	
R1-2	1560 (26%)	27 (12%)	
Surgical Procedure	*n* = 5480	*n* = 204	<0.001
PD	4045 (74%)	133 (65%)	
DP	773 (14%)	27 (13%)	
TP	653 (12%)	44 (22%)	
Disease Progression			
Local Recurrence	873/5974 (15%)	24/217 (11%)	0.011
Distant Metastases	2050/4451 (46%)	54/189 (29%)	<0.001

Pancreato-duodenectomy (PD); distal pancreatectomy (DP); total pancreatectomy (TP).

**Table 2 cancers-16-02016-t002:** Multivariable analysis using a Cox regression model for overall survival in the study cohort, *n* = 5503.

	Hazard Ratio	95% CI	*p*-Value
Age, >65 vs. <65 years	1.341	1.257–1.430	<0.001
Sex, Male vs. Female	1.097	1.034–1.164	0.002
Tumor Size (Reference = T1)			
T2 vs. T1	1.589	1.321–1.912	<0.001
T3–4 vs. T1	1.537	1.286–1.837	<0.001
Lymph Node Metastasis, N+ vs. N0	1.413	1.321–1.512	<0.001
Resection Margins, R+ vs. R0	1.463	1.369–1.564	<0.001
Histology, Invasive IPMN vs. PDAC	0.674	0.555–0.818	<0.001

*p*-value according to Cox regression analysis comparing the specified variables. HR indicates the hazard ratio.

## Data Availability

Data were obtained from the German Cancer Registry Group of the Society of German Tumor Centers and are available upon request from and under the regulations of the Society of German Tumor Centers.

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
