# Peer review of "Oncological Outcomes and Patterns of Recurrence after the Surgical Resection of an Invasive Intraductal Papillary Mucinous Neoplasm versus Primary Pancreatic Ductal Adenocarcinoma: An Analysis from the German Cancer Registry Group of the Society of German Tumor Centers"

_cancers, 2024, doi:10.3390/cancers16112016_

Round 1

Reviewer 1 Report (Previous Reviewer 3)

Comments and Suggestions for Authors

The authors addressed previously raised concerns.

Author Response

We would like to thank the Reviewer for his positive feedback regarding our study. 

Reviewer 2 Report (Previous Reviewer 2)

Comments and Suggestions for Authors

In comparision to previous version of the manuscript, the authors improved the quality of the article and changed its content accroding to the suggestions.

Author Response

We would like to thank the Reviewer for his positive feedback regarding our study. 

This manuscript is a resubmission of an earlier submission. The following is a list of the peer review reports and author responses from that submission.

Round 1

Reviewer 1 Report

Comments and Suggestions for Authors

This is a clinicopathological study to compare invasive IPMN with conventional PDAC using a large cohort.

The differences between conventional PDAC and invasive IPMNs have been studied many times, most revealing similar results; the clinicopathologic features of invasive IPMNs are relatively less malignant compared to conventional PDAC. However, the results regarding patient outcomes between these tumors are controversial. In addition, these differences were often observed early but not late in life. These two tumors also have different carcinogenic pathways. An open question is whether these two tumors have similar malignant performance. These backgrounds suggest that new studies should compare clinicopathologic characteristics, including patient survival, especially at each tumor stage.

The results of this study are mostly confirmative; the major problem is its study design. Patients with T1 and T2 tumors have different viability, and the T1/T2 ratios in this study cohort differ greatly between conventional PDAC and invasive IPMNs; T1 and T2 tumors are 3% and 30%, respectively, in PDAC and 22% and 30%, respectively, in invasive IPMNs. However, the authors performed survival analysis for T1 and T2 tumors together as a single category. This study design seemed inadequate to address the unresolved issues in this area of research. Even if patient outcomes differ at the T1/T2 stage, as the results of this study indicate, it is difficult to interpret whether this is due to differences in the biology of the two tumors or to differences in the stage of the invasive cancer.

Author Response

'Patients with T1 and T2 tumors have different viability, and the T1/T2 ratios in this study cohort differ greatly between conventional PDAC and invasive IPMNs; T1 and T2 tumors are 3% and 30%, respectively, in PDAC and 22% and 30%, respectively, in invasive IPMNs. However, the authors performed survival analysis for T1 and T2 tumors together as a single category.'

Thank you very much for your comment. We have decided to merge T1 and T2 tumors based on nodal status into two distinct groups: T1-T2 N0 or T1-T2 N+. This decision was driven by the observation that patients with T1 and T2 invasive IPMNs exhibited comparable median overall survival (OS) durations (39 and 34 months, p=0.995) compared to a median OS of 20 months in T3-4 tumors (T1 vs T3-4 p=0.018, T2 vs T3-T4 p=0.028, respectively). 

Additionally, patients with T1 N0 and T2 N0 invasive IPMN demonstrated similar OS rates (p=0.807) (refer to Figure S1 A and B). Moreover, the limited number of T1 N+ patients (n=3) does not offer sufficient statistical power for survival analysis. Despite these limitations, our study's substantial patient cohort and the distinct metastatic patterns observed between invasive IPMN and PDAC in early stages suggest different biological behaviors between these tumor types. (Figure S1 A and B)

Reviewer 2 Report

Comments and Suggestions for Authors

The article shows interesting data. However, it is not novel.

The sentence in the paragraph of 58-60: "In the absence...." is not true accoridng to European guidelines. If there are relative indications, surgery is recommended in some cases (not only absolute indications).

Why do you write that invasive IPMNs are associated with only lung metastases?

The title is not adeqaute to the content of article. When I red the title, I thought that the article demonstrates the problem of decision of surgery. A lot of cases of tumour of pancreas is unnecessarily resected. 

The results of your work are predictable. Results are not novel and explorative. They might be provided prior publication

Comments on the Quality of English Language

Some sentences require English correction

Author Response

'The sentence in the paragraph of 58-60: "In the absence...." is not true accoridng to European guidelines. If there are relative indications, surgery is recommended in some cases (not only absolute indications).'

Thank you very much for your comments. We corrected this sentence accordingly 

 …Currently, surgical resection is indicated in presence of certain high-risk stigmata, such as jaundice, enhancing mural nodules ≥ 5 mm, presence of solid components, positive cytology for HGD or invasive cancer and dilation of the pancreatic duct ≥ 10 mm (3, 5). The European guidelines further defined “worrisome features”, representing relative indications for surgery. Alternatively, patients with IPMN can be eligible for clinical surveillance using MRI and/or endoscopic ultrasound (EUS) technologies (4). Thereby, the surveillance intervals are dependent on the presence or absence of these worrisome features (3, 5)….  

'Why do you write that invasive IPMNs are associated with only lung metastases?'

Thank you for your input, we divided the metastasis pattern to lung-only metastases, liver-only metastases, peritoneal-only metastases etc. Invasive IPMN was in general associated with more lung-only metastases compared to PDAC (26% vs 14%). In early tumor stages (T1 and T2), patients were more likely to develop lung-only metastases compared to PDAC (75% and 36% vs 14% and 15%, p<0.001, respectively. To further clarify this, we added numbers to our graphs in (Figure 3).

Reviewer 3 Report

Comments and Suggestions for Authors

There are two major concerns in this study. These concerns may significantly affect the results of this study. The authors should provide reasonable solutions for these concerns.

1.     As authors stated, T staging was changed radically on 2016 when UICC 8th TNM system was published. However, this changing affects all T stages, not only T3-4 as they stated, because T staging was determined by tumor extent before 2016 but by just size after 2016. For example, T1 or T2 judged by 8th system (after 2016) could be categorized T3 or even T4 in the system before 2016. Therefore, T staging before and after 2016 could result in significant inconsistency.

2.     Another concern is how the size of tumor was determined in invasive IPMN. In 2015, a guideline for handling resected IPMN tissues was published; Adsay et al. Pathologic evaluation and reporting of intraductal papillary mucinous neoplasms of the pancreas and other tumoral intraepithelial neoplasms of pancreatobiliary tract: Recommendations of Verona consensus meeting. Ann Surg 263(1):162-77, 2016 (PMID 25775066). In this guideline, how to measure the size of invasive IPMN that is associated with T staging was clearly defined as measuring only invasive component, not IPMN itself represented by cyst size, which suggests that T staging depending on size had not been standardized before 2015.

Author Response

'There are two major concerns in this study. These concerns may significantly affect the results of this study. The authors should provide reasonable solutions for these concerns.

1.    As authors stated, T staging was changed radically on 2016 when UICC 8th TNM system was published. However, this changing affects all T stages, not only T3-4 as they stated, because T staging was determined by tumor extent before 2016 but by just size after 2016. For example, T1 or T2 judged by 8th system (after 2016) could be categorized T3 or even T4 in the system before 2016. Therefore, T staging before and after 2016 could result in significant inconsistency.'

Thank you for your invaluable input. We recognize the substantial alterations in T staging criteria in 2016; nevertheless, the fundamental definitions of T1 and T2 tumors have remained largely consistent. T1 and T2 tumors primarily represent smaller lesions, while T3 and T4 tumors typically encompass larger masses exceeding 4 cm, often extending beyond the confines of the pancreas.

Due to of the absence of metric data in our cohort, we decided to merge the T stages (T1/T2 and T3/T4) with nodal status to categorize tumors not only based on local extent but also considering their metastatic behavior. In this revised version, we conducted a sensitivity analysis of survival trends before and after 2016. Our findings demonstrate similar survival outcomes between the groups both pre- and post-2016, suggesting that while this grouping may lack granularity, it supports the robustness of our conclusions. (Figure S2).

'2.    Another concern is how the size of tumor was determined in invasive IPMN. In 2015, a guideline for handling resected IPMN tissues was published; Adsay et al. Pathologic evaluation and reporting of intraductal papillary mucinous neoplasms of the pancreas and other tumoral intraepithelial neoplasms of pancreatobiliary tract: Recommendations of Verona consensus meeting. Ann Surg 263(1):162-77, 2016 (PMID 25775066). In this guideline, how to measure the size of invasive IPMN that is associated with T staging was clearly defined as measuring only invasive component, not IPMN itself represented by cyst size, which suggests that T staging depending on size had not been standardized before 2015.'

We value your input. Tumor size was determined by pathologists from multiple institutions and categorized according to the ICDO3 system based on WHO definitions available at the time of assessment. Additionally, we conducted a sensitivity analysis to examine survival trends specifically for IPMNs before and after 2016, akin to the example provided earlier. We acknowledge that post-2016, the difference in survival became more pronounced compared to pre-2016. However, our pooled analysis revealed that both time periods retained statistical significance (p=0.047 vs. p<0.001). A subgroup analysis was not conducted due to the limited number of patients within the subgroups, which would yield inconclusive results. We have added this limitation into our study (Figure S3). 

Reviewer 4 Report

Comments and Suggestions for Authors

The clinical outcome of patients resected due to invasive IPMN is a very interesting subject, that is analyzed increasingly. Therefor this manuscript is of clinical interest, although there are shortcomings in data presentation and discussion.

Registry data from multiple institutions may be problematic - were the data audited?

How often was the diagnosis of invasive IPMN settled preoperatively? Or was the majority diagnosed from resected specimen after pathology accidentally?

The evidence for a difference of the biological course of invasive IPMN and PDAC is not high. One could argue, that more advanced stages of invasive IPMN are not different from PDAC. Is there a possible difference only in early tutor stages and is it really a different biology or more a selection in the time point of diagnosis? This has to be addressed in the discussion.

Why did less than 50% with invasive IPMN did not receive any chemotherapy? Only a minority with invasive IPMN received a standard regimen? Was this institution dependent or what else?

Conclusions regarding a comparison of the effect of adjuvant chemotherapy in invasive IPMN patients are not valid due to this insufficient data concerning  the proportion of treated patients and the choice of the chemotherapeutic regimen. This must be addressed as major shortcoming in the discussion. 

Author Response

'1.    Registry data from multiple institutions may be problematic - were the data audited?'

Thank you for your comment. The data presented here originates from 24 distinct clinical cancer registries, implying an anticipated degree of variability in data acquisition. Moreover, the available dataset is not specific for invasive IPMNs or PDAC but rather generalised for C.25 tumors ( according to ICDO3 WHO Classification 2018). The German Cancer Registry Group of the Society of German Tumor Centers (ADT) has implemented training programs aimed at standardizing data input procedures, involving a 20-day training period followed by examination under supervision, as detailed in: 

( https://www.adt-netzwerk.de/Fortbildungen/Fortbildungsreihe_und_Zertifikat_Tumordokumentar_in/).

This limitation has been previously mentioned in our limitation part. 

'2.    How often was the diagnosis of invasive IPMN settled preoperatively? Or was the majority diagnosed from resected specimen after pathology accidentally?'

This information is not available in our dataset. However, the high incidence of small tumors in IPMN is suggestive that IPMNs were mostly under surveillance. 

'3.    The evidence for a difference of the biological course of invasive IPMN and PDAC is not high. One could argue, that more advanced stages of invasive IPMN are not different from PDAC. Is there a possible difference only in early tutor stages and is it really a different biology or more a selection in the time point of diagnosis? This has to be addressed in the discussion.'

Thank you for your input. We agree with your conclusion, according to our results, the main difference in survival between PDAC and invasive IPMN is in localized disease (T1-T2 N0) with less pronounced effect in higher tumor stages (Figure 2).

'4.    Why did less than 50% with invasive IPMN did not receive any chemotherapy? Only a minority with invasive IPMN received a standard regimen? Was this institution dependent or what else? Conclusions regarding a comparison of the effect of adjuvant chemotherapy in invasive IPMN patients are not valid due to this insufficient data concerning  the proportion of treated patients and the choice of the chemotherapeutic regimen. This must be addressed as major shortcoming in the discussion.'

The exact reasons for the omission of adjuvant chemotherapy are not documented in our dataset. In a study conducted by Petruch et al., comparing standard outcomes in resectable pancreatic cancer between Germany and the US, it was found that only 36% of resected pancreatic cancer patients in Germany and 30% in the US received adjuvant chemotherapy. Hence, while the numbers are relatively low, they are reflective of real-world practices (DOI: 10.1016/j.surg.2023.11.004). We added this point to our discussion.

Reviewer 5 Report

Comments and Suggestions for Authors

In the manuscript titled “Oncological Outcomes and Patterns of Recurrence After Surgical Resection of Invasive Intraductal Papillary Mucinous Neoplasm versus Primary Pancreatic Ductal Adenocarcinoma. An analysis from the German Cancer Registry Group of the Society of German Tumor Centers” a team of researchers from Germany present a large retrospective study comparing outcomes of surgical treatment of invasive IPMN and primary PDAC. Study is based on many patients and the presented data is interesting. However, I have several comments and questions:

1.     After reading the manuscript for me it is still not totally clear what is the clinical relevance of the presented data and conclusions

2.     5794 patients with PDAC are compared to 217 patients with IPMN. Only 40% patients with invasive IPMN received chemotherapy and majority of them received gemcitabine. Were both cohorts just compared or also matched? Matching the cohorts might influence the outcomes of the study.

Author Response

'In the manuscript titled “Oncological Outcomes and Patterns of Recurrence After Surgical Resection of Invasive Intraductal Papillary Mucinous Neoplasm versus Primary Pancreatic Ductal Adenocarcinoma. An analysis from the German Cancer Registry Group of the Society of German Tumor Centers” a team of researchers from Germany present a large retrospective study comparing outcomes of surgical treatment of invasive IPMN and primary PDAC. Study is based on many patients and the presented data is interesting. However, I have several comments and questions:

1. After reading the manuscript for me it is still not totally clear what is the clinical relevance of the presented data and conclusions.'

Thank you for this comment. Here, we present the first study from Germany regarding patient outcomes after resection of invasive IPMNs compared to PDAC. Our data shows that small invasive IPMNs are associated with greater survival outcomes compared to later stages, emphasizing on the importance of a low threshold on tumor resection in patients under surveillance. Furthermore, we give an insight on the different metastatic patterns in IPMNs and finally we discuss the effect of adjuvant chemotherapy in invasive IPMNs.

'2. 5794 patients with PDAC are compared to 217 patients with IPMN. Only 40% patients with invasive IPMN received chemotherapy and majority of them received gemcitabine. Were both cohorts just compared or also matched? Matching the cohorts might influence the outcomes of the study.'

Thank you for sharing your insights. Propensity score matching (PSM) serves as an alternative comparison technique. However, a significant drawback of PSM is the potential loss of information concerning the biological and metastatic patterns of unmatched patients in the analysis. As an alternative approach, we performed multivariable regression analysis, which identifies independent prognostic factors while retaining a larger dataset, enabling more detailed statistical analysis.